# Hypolipidemic Effect of *Arthrospira* (*Spirulina*) *maxima* Supplementation and a Systematic Physical Exercise Program in Overweight and Obese Men: A Double-Blind, Randomized, and Crossover Controlled Trial

**DOI:** 10.3390/md17050270

**Published:** 2019-05-07

**Authors:** Marco Antonio Hernández-Lepe, Abraham Wall-Medrano, José Alberto López-Díaz, Marco Antonio Juárez-Oropeza, Oscar Iván Luqueño-Bocardo, Rosa Patricia Hernández-Torres, Arnulfo Ramos-Jiménez

**Affiliations:** 1Medicine and Psychology School, Autonomous University of Baja California, Tijuana 22390, Mexico; marco.antonio.hernandez.lepe@uabc.edu.mx; 2Biomedical Sciences Institute. Autonomous University of Ciudad Juarez, Ciudad Juarez 32310, Mexico; awall@uacj.mx (A.W.-M.); joslopez@uacj.mx (J.A.L.-D.); 3Medicine School, National Autonomous University of Mexico, Mexico City 04510, Mexico; majo_ya@yahoo.com.mx (M.A.J.-O.); luqueno@bq.unam.mx (O.I.L.-B); 4Physical Culture Sciences School, Autonomous University of Chihuahua, Chihuahua 32310, Mexico; rphernant@yahoo.com

**Keywords:** *Arthrospira maxima*, dyslipidemia, physical exercise, obesity, double-blind, randomized controlled trial

## Abstract

Low-fat diets, lipid-modifying nutraceuticals and a higher level of physical activity are often recommended to reduce dyslipidemia. A double-blind, randomized, crossover, controlled trial was designed to evaluate the independent and synergistic effects of *Arthrospira* (*Spirulina*) *maxima* supplementation (4.5 g·day^−1^) with or without performing a physical exercise program (*PEP*: aerobic exercise (3 days·week^−1^) + high-intensity interval training (2 days·week^−1^)) on blood lipids and BMI of 52 sedentary men with excess body weight. During six weeks, all participants were assigned to four intervention treatments (*Spirulina maxima* with PEP (SE), placebo with PEP (Ex), *Spirulina maxima* without PEP (Sm), placebo without PEP (C; control)) and plasma lipids were evaluated spectrophotometrically pre- vs. post intervention in stratified subgroups (overweight, obese and dyslipidemic subjects). Pre/post comparisons showed significant reductions in all plasma lipids in the SE group, particularly in those with dyslipidemia (*p* ≤ 0.043). Comparing the final vs. the initial values, BMI, total cholesterol, triglycerides and low-density lipoprotein cholesterol were decreased. High-density lipoprotein cholesterol increased in all treatment groups compared to C. Changes were observed mostly in SE interventions, particularly in dyslipidemic subjects (*p* < 0.05). *Spirulina maxima* supplementation enhances the hypolipidemic effect of a systematic PEP in men with excess body weight and dyslipidemia.

## 1. Introduction

Dyslipidemia is an abnormal clinical condition characterized by the altered level of one or more plasma lipids, including but not restricted to total (TC), low-density lipoprotein (LDL-C) and high-density lipoprotein (HDL-C) cholesterol and triglycerides (TG). Dyslipidemias often increase concomitantly with body mass index (BMI) and central adiposity, increasing the risk for metabolic syndrome and cardiovascular diseases (CVD) [1]; its pathophysiology is multifactorial but excess of body weight and sedentarism are two of the most important factors [2]. Effective control measures to stop the pandemic obesity-dyslipidemia consortium are a priority for health systems worldwide [3].

Lifestyle changes aimed to reduce people’s sedentarism and the promotion of healthy eating patterns have been widely used to prevent and even treat obesity and dyslipidemias [4,5]. However, moderate physical activity alone is not effective to lose weight or body fat [6] but systematic exercise programs based in high-intensity protocols reduces CVD risk and dyslipidemia [7]. A heart-healthy diet aimed to reduce the intake of saturated/ trans fatty acids and cholesterol, due to their unfavorable metabolic fate, must include functional foods (e.g., plant and marine proteins) and nutraceuticals (e.g., *n*-3 fatty acids, phytosterols and polyphenols) [8,9]. *Arthrospira maxima*, commercially known as *Spirulina maxima* (*S. maxima*), is a cyanobacteria used as a nutritional supplement due to its high content of protein, essential fatty acids, vitamins, polyphenols, carotenoids and phycocyanins [10] with cardio-protective and antioxidant activity. In this regard, Moura et al. [11] reported that *Spirulina* supplementation decreases circulating LDL-C levels more effectively than aerobic training in diabetic Wistar rats while *Spirulina* intake and aerobic physical exercise resulted in even better effects. Mazzola et al. [12] demonstrated that *Spirulina* and physical reduces plasma TG levels in Wistar rats Kata et al. [13] reported reduced levels of TC, TG and LDL-C in hypercholesterolemic rabbits. However, these studies were conducted in animal models and therefore their results cannot be extrapolated to humans, and clinical studies conducted in humans are still very scarce [14].

This study aimed to assess the independent and synergistic effect of *S. maxima* supplementation (4.5 g·day^−1^) and the practice of systematic physical exercise on plasma lipid levels in young subjects with overweight, obesity or defined dyslipidemia. We hypothesize that *S. maxima* intake with or without a systematic physical exercise program will decrease the BMI while improving plasma lipids in male patients with overweight and obesity. 

## 2. Results

Fifty-two male young (26 ± 5 years) subjects with excess weight (BMI ≥ 25 kg·m^−2^) were enrolled in the study (Table 1). 

Baseline characteristics of subjects at day 0 and 56 (wash-out period) did not show differences (*p* < 0.05; see Appendix A). Of all the participants, 52 and 48% were overweight and obese, respectively, while dyslipidemic subjects were distributed as follow: High TC (7 overweight, 8 obese), high TG (11 overweight, 16 obese), high LDL-C (17 overweight, 21 obese) and low HDL-C (17 overweight, 18 obese). 

### 2.1. Diet

The daily energy intake after the six weeks of supplementation showed no statistical differences (*p* < 0.05) for dietary variables at the beginning (2054 ± 104 kcal·day^−1^) compared with those at the end of the study (2146 ± 98 kcal·day^−1^). No adverse effects of dietary or *S. maxima* supplementation were reported during the study. 

### 2.2. Intra Group Comparisons (Pre vs. Post) on the Blood Lipid Profile

After 42 days of treatment, there was an improvement (*p* < 0.05) between basal (pre) and final (post) levels of TC, LDL-C and HDL-C in the SE intervention, and by Ex only in LDC-C considering all participants (overweight + obesity, Table 2a). Analysing dyslipidemic participants only, SE presented an improvement for all analysed parameters of blood lipids (*p* < 0.05), Ex on TC, LDL-C and HDL-C, and Sm on TC, TG and HDL-C (Table 2b).

### 2.3. Inter-Group Comparisons of the Blood Lipid Profile

To achieve an objective comparison of the effects of each intervention, absolute changes were calculated (Δ change = *Post* − *Pre* values) for TC and TG (Figure 1; mg·dL^−1^), LDL-C and HDL-C (Figure 2; mg·dL^−1^) and BMI (Figure 3; kg·m^−2^). As compared to C (placebo+no exercise), important reductions in plasma TC levels (Δ change, mg.dL^−1^ [upper; lower quartile]) were observed in overweight (−24 [−43; −17], −17 [−22; −8]) obese (−34 [−44; −24], −23 [−34; −13]) and dyslipidemic (−34 [−43; −29], −25 [−35; −14]) subjects enrolled in SE and Ex. Also, a mild yet significant (*p* < 0.05) reduction in TC level was observed in obese subjects in the Sm treatment (−24 [−34; −18]) and TG level in overweight (−26 [−36; −17]) subjects enrolled in the SE treatment.

As compared to C, the LDL-C plasma level (Δ change; mg·dL^−1^; [upper; lower quartile] Figure 2) in SE, Ex and Sm was lower in overweight (SE (−26 [−49; −19]) > Ex (−19 [−26; −13]) and Sm (−19 [−24; −12])), obese (SE (−33 [−63; −20]) > Ex (−30 [−40; −17]) > Sm (−21 [−31; −17])), and dyslipidemic (SE (−37 [−55; −26]) > Ex (−25 [−30; −16]) > Sm (−19 [−24; −16])) subgroups (*p* < 0.05). On the other hand, HDL−C levels (Δ change, mg·dL^−1^ [upper; lower quartile]) only improved in overweight (+7, [4; 11]) and dyslipidemic (+8, [4; 13]) subjects from the SE treatment.

### 2.4. Inter-Group Comparisons on the Body Mass Index

BMI reduction (Δ change, kg·m^−2^; [upper; lower quartile]; Figure 3) was only observed in overweight (−0.6 [−1.2; −0.60]; −0.4 [−0.7; −0.3]) and dyslipidemic (−0.7 [−1.3; −0.4]; −0.4 [−0.7; −0.3]) subjects enrolled in SE and Sm treatments. Such reductions in BMI directly correlated with TC (*r* = 0.49; *p* < 0.01), TG (*r* = 0.22; *p* < 0.05), and LDL-C (*r* = 0.42; *p* < 0.01).

## 3. Discussion

The excess of body weight is highly associated with dyslipidemias [15]; the world health care systems yearly boost new initiatives to prevent and treat these and other CVD risks [16]. This study provides unique evidence on the synergistic effects of *S. maxima* supplementation (4.5 g·day^−1^) and a systematic exercise program on improving blood lipid levels in overweight, obese and dyslipidemic subjects by using a double-blind, randomized, crossover trial design. It is noteworthy that very few studies have been focused on *Spirulina* supplementation on dyslipidemia. However, even without being part of the hypothesis, the main finding of this study was the improvement on blood lipids concentration of dyslipidemic men by six weeks of treatment with *S. maxima* supplementation (4.5 g·day^−1^) and/or systematic physical exercise practice.

There is no information in the literature about clinical trials using *Spirulina* together with aerobic exercise or high-intensity interval training. Only a few studies have focused on *Spirulina* supplementation effects against dyslipidemia and CVD risk factors. Mani et al. [17] studied the effect of *Spirulina* supplementation (2 g·day^−1^) during two months on the serum lipid profile of 15 patients affected by type II diabetes mellitus (T2DM), resulting in a significant reduction of TG, TC, LDL-C, and free fatty acid in blood concentrations. By means of a better-structured trial, Lee et al. [18] studied the effect of *Spirulina* supplementation (8 g·day^−1^) during 12 weeks in 37 patients with T2DM, reporting a significant reduction in TG levels after the intervention.

Many of the beneficial effects of *Spirulina* are attributed to its nutritional content, but its action mechanisms are poorly understood [19]. Our findings corroborate previous investigations in animal models. Some authors suggest that a possible component responsible of the *S. maxima* hypolipidemic effect is C-phycocyanin protein, which improves the blood lipid profile. Iwata et al. [20] studied the effect of a diet containing *Spirulina* compared to a high fructose diet in rats; they reported a decrease in TG concentration after the intervention, and attribute the possible action mechanism to the lipoprotein lipase activity in the lipoprotein metabolism. Other authors suggest that C-phycocyanin increases endogenous enzymes activity, scavenging free radicals [21], and downregulates cofactors in fat metabolism like adenine dinucleotide phosphate [22]. Nagaoka et al. [23] attribute *Spirulina* hypolipidemic effects to the fact that dietary supplementation with the cyanobacteria seems to have decreased the intestinal assimilation of cholesterol, probably because *Spirulina* compounds bind to bile acids in the jejunum, affecting the micellar solubility of cholesterol before suppressing the cholesterol absorption. These biological processes might explain the underlying mechanisms of *S. maxima* involved in the significant improvement on TC, TG, and lipoprotein-associated cholesterol found in this research, but appropriate clinical trials are needed to elucidate this. 

Recent studies focus mainly on controlling serum LDL-C concentrations because this lipid is the primary carrier of cholesterol in the blood (60–80% of TC), and when this lipoprotein is decreased, blood TC is also decreased [24]. Although drugs exist against dyslipidemia, they are associated with adverse effects like mild serum creatine kinase level elevations and skeletal muscle complaints, including rhabdomyolysis and myopathy [25], reasons why different alternative treatments are investigated. The main treatment of dyslipidemia is an adequate dietary therapy, including nutraceuticals or functional foods [26], but it is essential to consider all possible factors that trigger this disease, like a sedentary lifestyle.

Actual research focused on physical activity has found that the benefits of aerobic exercise are due to changes in the physical structure of blood-carrier cholesterol proteins, resulting in an improvement of lipid levels [27]. Specifically, systematic exercise increases the activity of lipoprotein lipase that hydrolyzes blood TG, then it acts on lipoprotein particles through the capillaries, releasing free fatty acids that may be taken up by skeletal muscle [28]. Nikolaidis et al. [29] suggest that this degradation of endogenous lipids causes a shrinkage of lipoprotein particles, inducing the transference of lipids between very low-density lipoprotein and the HDL-C. 

Tan et al. [30], in a randomized trial, studied the effect of 10 weeks of supervised aerobic exercise (five days per week, one hour per day) in 30 overweight women, resulting in a decrease in body weight, BMI, body fat, and an improvement in the blood lipid profile. Cugusi et al. [31], in an observational study, reported the effect of 12 weeks of an exercise program (50 min a day, 3 days a week) in 18 men affected by type II diabetes mellitus; they report a reduction of TC, TG, and LDL-C. These studies and others [32] support the conclusion that aerobic exercise reduces that blood lipid levels mostly when there exists a decrease in body weight. We previously reported [33] a positive effect of physical exercise and *S. maxima* supplementation on the reduction of body weight and body fat. Type II diabetes mellitus, sedentarism, dyslipidemia, and overweight are all related, a reason why a beneficial effect for any of them can be related to the other ones [34].

The greater synergistic effect of *S. maxima* and systematic physical exercise observed in the obesity group compared with the overweight one was not only due to their elevated BMI and to their dyslipidemia problem. The fact that the participants with obesity that achieved a reduction in BMI showed a difference only in SE compared with control group is not surprising, given that in obesity there exists a lower activity of oxidative enzymes, resulting in an elevated intramuscular lipid content, mostly TG [35]. For that reason, the capacity of obese subjects to use TG-derived fatty acids and circulating LDL-C as fuel during physical activity resulted in a higher lipid oxidation, which can be corroborated with the higher decrease of the blood lipid profile and the BMI in obesity [36].

The individual differences were wide-ranging due to different factors, like genetics [37] or gut microbiota [38]; consequently, it is essential to clarify that our treatment could not be sufficient for reducing cardiovascular risks, but trials of long duration or in different kind of populations can be conducted, measuring not only the blood lipids, but specific markers like creatine kinase, glucose or enzymatic activity to understand better the action mechanism of exercise and/or *S. maxima* intake.

The strong points of the present study were: no missing data, no participants dropped out during the trial, and the double-blind randomized protocol. The beneficial results suggest a synergistic effect of systematic physical exercise and *S. maxima*, resulting in an improvement of the blood lipid profile. Limitations of the study were that only sedentary overweight and obese men were selected, so the results may not be the same in other populations.

## 4. Materials and Methods

*Spirulina maxima* was obtained commercially from Alimentos Esenciales para la Humanidad S.A. de C.V. (Mexico City, México) in January 2017, and its chemical/functional characterization and biosafety were evaluated before being used in this clinical trial [33].

### 4.1. Ethics

The CONSORT checklist (Appendix A) and flow diagram (Appendix A) are available as Appendix A. All participants were informed of the study purpose, physical, clinical and biochemical procedures. Their acceptance was formalized through informed consent, and their anonymity and confidentiality were strictly enforced. The clinical procedures were previously described at ClinicalTrials.gov with trial registration number NCT02837666 (Hypolipidemic and Antioxidant Capacity of *Spirulina* and Exercise Registered 19 July 2016 https://clinicaltrials.gov/ct2/show/NCT02837666?cond=Hypolipidemic+and+Antioxidant+Capacity+of+Spirulina+and+Exercise&rank=1). A detailed protocol for this trial has been published previously [39]. This clinical trial was carried out in accordance with the declaration of Helsinki and was approved by Universidad Autónoma de Ciudad Juárez (UACJ) Review Board: (Reference number: CBE.ICB/062.09-15).

### 4.2. Participants’ Eligibility

Fifty-two sedentary male adults with a BMI over 25 kg·m^−2^ (27 overweight and 25 obese) volunteered to participate from May to September 2017. For the recruitment of participants, an intra-school campaign and personalized interviews were conducted to ensure eligibility. The exclusion criteria of subjects were these: drinking more than 100 mL of alcohol a week, taking drugs and/or diet supplements, presenting chronic disease, and having an impediment to practicing regular physical exercise. The elimination criteria were attendance by the subject of < 80% to the physical exercise sessions. No participants dropped out of the study.

### 4.3. Baseline Measurements

Initially, each participant visited the exercise physiology laboratory at UACJ with a fasting of 8–10 h before the baseline evaluations. BMI measurements were performed with subjects lightly dressed and barefoot, using an electronic balance (SECA 876, Hamburg, Germany) and the standing height was measured with a stadiometer (SECA 206, Hamburg, Germany). Participants’ blood sample was taken by an expert clinician from the antecubital vein.

### 4.4. Study Design

The clinical trial consisted of *S. maxima* (4.5 g·day^−1^) or placebo (4.5 g·day^−1^ of a low-calorie saccharine powder) supplementation (both supplements were encapsulated in dark capsules to present the same organoleptic characteristics) during 12 weeks in a randomized, double-blind, placebo-controlled, and counterbalanced crossover trial in a 2 × 2 factorial design.

To avoid study desertion, eligible participants (*N* = 52) decided whether or not to participate in the systematic physical exercise program (where they stayed during the two treatments of the clinical trial). Then, they were randomly allocated to one of two possible supplementation interventions (*S. maxima* or placebo) divided in four treatments, including physical exercise program with *S. maxima* (SE) supplementation, physical exercise with placebo (Ex) supplementation, *S. maxima* supplementation without physical exercise program (Sm), or placebo supplementation without physical exercise program (C). The crossover was conducted for the supplementation interventions and to assess compliance to the supplement intake. Participants returned each week to the laboratory to receive new capsules and all treatment intake data were recorded on weekly case report forms.

The initial allocation was performed in such way that each group had almost the same proportion of overweight: obese (1:1) individuals using a computer-generated random schedule, stratified by a permuted block design because stratification in small trials confers an adequate balance and slightly more statistical power and precision [40]. Participants’ group allocations were performed by an independent researcher, who did not have any other participation during the study (double-blind). The sample size was determined by using the statistical program G*Power [41], selecting a sample of >52 subjects, with α = 0.05 and power = 0.85.

The first period of treatment was carried out for six weeks, followed by a two-week wash-out period to avoid any possible delayed effect of *S. maxima* in the organism, and, finally, a further six weeks of treatment for the second intervention groups (Figure 4). Due to the lack of information in the literature, the durations for both the wash-out period (2 weeks) and treatment period (6 weeks) were considered long enough according to a systematic review of clinical trials that used *Spirulina* as treatment [42].

### 4.5. Blood Sample Collection and Biochemical Analysis

Four blood samples were collected during the clinical trial, first on day 0, second on the final of first treatment (day 42), third after the washout period (day 56), and fourth at the end of the second treatment (day 96). Blood samples (8 mL) were collected from the antecubital vein into ethylene diamine tetra acetic acid (EDTA) tubes after 8–10 h of fasting. Plasma from blood samples was obtained by refrigerated centrifugation (4 °C) at 3,000 g for 20 min. Plasma lipid profile (TC, HDL-C, and TG) concentrations were analyzed by using standard enzymatic procedures (Spinreact, Girona, Spain) with a microplate spectrophotometer (Epoch, Biotek, VT, USA) at 505 nm, and LDL-C was calculated with Friedewald’s formula [43]. Abnormal cutoffs for TC, TG, LDL-C and HDL-C were >200, >150, >100, and <40 mg·dL^−1^, respectively.

### 4.6. Dietary Analysis

All participants were subjected to a nutritional survey to define the daily calories required to establish dietary recommendations. Dietary intake was monitored by two trained nutritionists using three 24 h dietary recalls (semi-quantitative method), on days 0, 42, 56, and 98, and two food frequency questionnaires (qualitative method), on days 0 and 98, according to Nelson’s guidelines [44], in order to ensure the independence of both dietary assessment tools. Dietary records were first inspected for missing data (e.g., missing food items or no complete responses) by a nutritionist and then analyzed for total calories, protein, carbohydrate, and fat intake (Diet Analysis Plus, ESHA Research, Salem, OR, USA). Lastly, compliance with the intake of supplements (*S. maxima* or placebo) and diet was monitored weekly by scheduled laboratory visits and carried out by trained nutritionists.

### 4.7. Systematic Physical Exercise Protocol

Exercise prescription was individual and accorded with the American College of Sports Medicine (ACSM) recommendations for people with overweight and obesity [45]. Participants in SE and Ex exercised for five days a week. The protocol began with a warm-up exercise (5–10 min), followed by muscular endurance exercise (20–30 min), then 20–30 min of cardiovascular exercise (walking, jogging, running, or cycling), and finally, five minutes of stretching. The intensity of cardiovascular exercise was administered as follows: For three of the five days, the intensity was between 50% and 80% of their heart rate reserve, and for the other two days, the intensity was between 80% and 90% of their heart rate reserve, using high-intensity interval training. 

Muscular endurance exercise consisted of working all muscle groups (arms, legs, chest, back, and shoulders) once a week, using a medium-resistance repetition protocol divided in four different exercises for each specific muscle group, doing three sets of 12–16 repetitions [45]. The protocol for the high-intensity interval training consisted of 5 to 7 sets of 1 min of running at 80–90% heart rate reserve followed by 3 min of active resting at 50–70% of this heart rate. Heart rate was monitored since it is closely related to the percent of oxygen uptake reserve, making it easier to verify exercise intensity in the exercise prescription program [46]. Subjects performed the physical exercise program in the UACJ gym, always under the technical supervision of a personal trainer.

### 4.8. Main Physiological Outcomes

The primary outcome of the study was the response of treatments on the TC, TG, LDL-C, and HDL-C blood levels of dyslipidemic overweight and obese subjects, because high levels of blood lipids are of the major risk factors of CVD, and health organizations around the world recommend studies focused on improving these response variables [1].

We included as secondary endpoints the BMI and the blood lipid profile of all the participants of the study, since excess weight has been associated with CVD, regardless of whether dyslipidemia occurs [47].

### 4.9. Statistical Analysis

All analyses were conducted using the software SPSS 22.0 (SPSS Inc., Chicago, IL, USA) (Appendix A). Data distribution normality was examined by the Shapiro–Wilk test, and the homoscedasticity by the Levene test, for each group. A *p*-value of less than 0.05 was considered statistically significant. In order to analyze statistical differences among treatments and time, univariate repeated measures ANOVA design with two within-subjects (initial and final values), and four inter-subjects (treatments) factors was used. In addition, initial and final values were compared using paired t-tests. When the variables had a normal distribution (*p* > 0.05) and variances between groups were the same, the Δ was analyzed by one-way ANOVA and Tukey Post-hoc test, but when the variables did not have a normal distribution (*p* < 0.05) and variances between groups were different, the Δ was analyzed by Kruskal-Wallis H and Dunn’s Post-hoc test with Bonferroni adjustment for multiple comparisons. To evaluate the associations among variables, a Spearman correlation was performed. Data are presented as the mean ± standard deviation (SD) and, when specified, as the median with lower and upper quartiles. 

## 5. Conclusions

According to the results, *Spirulina maxima* supplementation enhances the effect of a short-term systematic physical exercise program on BMI and blood lipid profile observed in overweight and obese men, but mostly in individuals with dyslipidemia.

## Figures and Tables

**Figure 1 marinedrugs-17-00270-f001:**
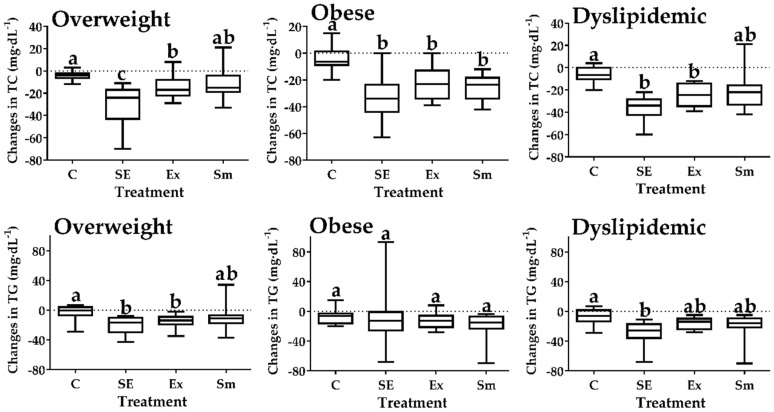
Changes (Δ) in total cholesterol and triglycerides by treatments. SE: *Spirulina* and exercise; Ex: Exercise and placebo; Sm: *Spirulina* without exercise; C: Control (Placebo without exercise); TC: Total cholesterol; TG: Triglycerides. Data presented as box and whisker plots of the median (horizontal line), upper and lower quartiles (box) maximum and minimum (error bars). Different letters indicate statistical difference between treatments (*p* < 0.05), Kruskal-Wallis Test with Dunn’s post-hoc for all panels of changes in TC and for changes in TG of overweight and dyslipidemic, ANOVA test for changes in TG of obese subjects.

**Figure 2 marinedrugs-17-00270-f002:**
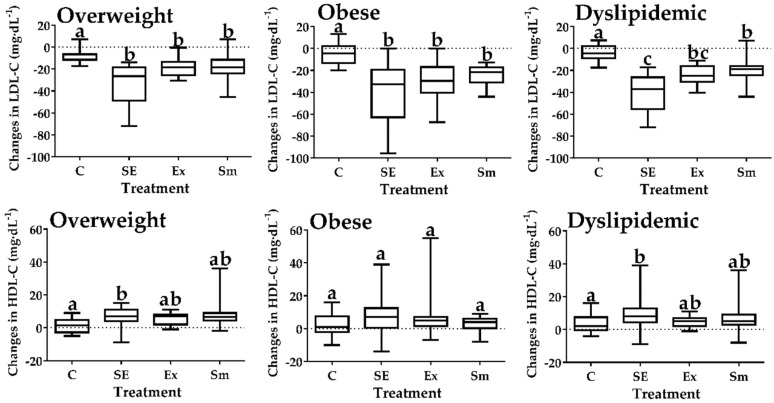
Changes (Δ) in lipoproteins levels by treatments. SE: *Spirulina* and exercise; Ex: Exercise and placebo; Sm: *Spirulina* without exercise; C: Control (Placebo without exercise); LDL-C: Low-density lipoprotein-cholesterol; HDL-C: High-density lipoprotein-cholesterol. Data presented as box and whisker plots of the median (horizontal line), upper and lower quartiles (box) maximum and minimum (error bars). Different letters indicate statistical difference between treatments (*p* < 0.05), Kruskal-Wallis Test with Dunn’s post-hoc for all panels of changes in LDL-C and for changes in HDL-C of overweight and dyslipidemic, ANOVA test for changes in HDL-C of obese subjects.

**Figure 3 marinedrugs-17-00270-f003:**
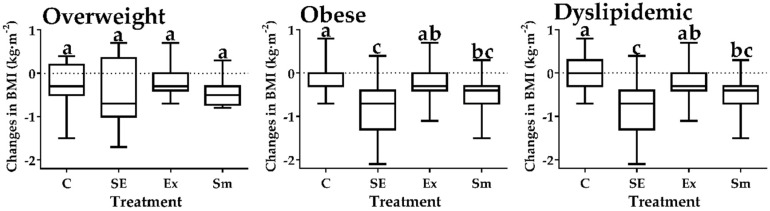
Changes (Δ) in body mass index by treatments. SE: Spirulina and exercise; Ex: Exercise and placebo; Sm: Spirulina without exercise; C: Control (Placebo without exercise); BMI: Body mass index. Data presented as box and whisker plots of the median (horizontal line), upper and lower quartiles (box) maximum and minimum (error bars). Different letters indicate statistical difference between treatments (*p* < 0.05), Kruskal-Wallis test with Dunn’s post-hoc for changes in BMI of obese and dyslipidemic, ANOVA test for changes in BMI of overweight subjects.

**Figure 4 marinedrugs-17-00270-f004:**
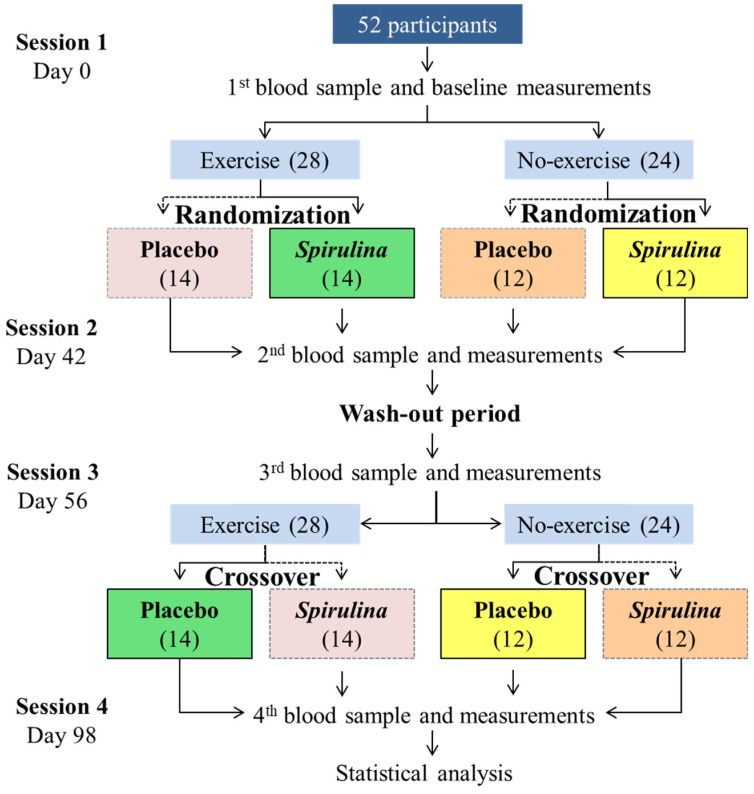
Experimental design for the independent and synergistic effect of *Spirulina maxima* and exercise. The same color means the same group of participants.

**Table 1 marinedrugs-17-00270-t001:** Baseline characteristics of participants.

	Total	Overweight	Obese
*n*	52	27	25
Age (years)	26 ± 5	26 ± 4	27 ± 6
Body weight (kg)	90 ± 13	81 ± 8	100 ± 12
Height (m)	1.72 ± 0.1	1.72 ± 0.1	1.73 ± 0.1
BMI (kg·m^−2^)	30.2 ± 4	27.4 ± 1.2	33.3 ± 3.8
Energy intake (kcal·day^−1^)	2054 ± 104	1977 ± 139	2054 ± 151
Total cholesterol (mg·dL^−1^) *	196 ± 36	177 ± 28	218 ± 30
Triglycerides (mg·dL^−1^)	142 ± 41	136 ± 35	150 ± 46
LDL-C (mg·dL^−1^) *	134 ± 36	115 ± 27	158 ± 31
HDL-C (mg·dL^−1^)	34 ± 9.6	35.3 ± 8.2	32.5 ± 10.9

Data are expressed as mean ± SD. Asterisk (*) means statistical differences comparing overweight and obese individuals (*p* < 0.05); *n*: Sample size, BMI: body mass index, LDL-C: low-density lipoprotein-cholesterol, HDL-C: high-density lipoprotein-cholesterol.

**Table 2 marinedrugs-17-00270-t002:** Effect of treatments on the blood lipid profile within the total and dyslipidemic participants.

**Blood lipid**		**2a (Overweight and obese subjects)**			
		**C**	**SE**	**Ex**	**Sm**
**TC**	Basal	190±31	196±35	197±38	201±39
	Final	186±31	163±33	177±36	183±37
	*p* value (*n*)	0.619 (12)	0.001 (14) *	0.053 (14)	0.103 (12)
**TG**	Basal	131±31	157±47	139±44	141±37
	Final	125±32	135±37	124±42	127±34
	*p* value (*n*)	0.517 (12)	0.070 (14)	0.212 (14)	0.176 (12)
**LDL-C**	Basal	131±32	128±36	137±38	138±37
	Final	127±30	93±34	115±36	117±38
	*p* value (*n*)	0.680 (12)	0.001 (14) *	0.046 (14) *	0.062 (12)
**HDL-C**	Basal	35±12	34±9	31±9	35±10
	Final	37±8	42±10	36±9	40±11
	*p* value (*n*)	0.655 (12)	0.005 (14) *	0.059 (14)	0.071 (12)
**Blood lipid**		**2b (Dyslipidemic subjects)**			
		**C**	**SE**	**Ex**	**Sm**
**TC**	Basal	219±16	226±22	232±23	233±21
	Final	213±18	189±20	208±28	212±23
	*p* value (*n*)	0.412 (6)	0.000 (7) *	0.025 (6) *	0.029 (6) *
**TG**	Basal	160±6	184±40	180±25	167±11
	Final	153±12	156±29	164±21	148±19
	*p* value (*n*)	0.156 (6)	0.043 (7) *	0.096 (7)	0.004 (7) *
**LDL-C**	Basal	140±29	141±29	148±33	148±33
	Final	135±27	101±34	124±33	128±32
	*p* value (*n*)	0.650 (9)	0.001 (10) *	0.030 (10) *	0.060 (9)
**HDL-C**	Basal	28±8	30±6	28±6	29±6
	Final	31±5	40±10	33±6	35±10
	*p* value (*n*)	0.172 (7)	0.001 (10) *	0.019 (10) *	0.040 (8) *

Data are expressed as mg·dL^−1^ and mean ± SD. Asterisk (*) means statistical differences between basal and final blood lipid concentration (*p* < 0.05). SE: *Spirulina* and exercise; Ex: Exercise and placebo; Sm: *Spirulina* without exercise; C: Control (Placebo without exercise); TC: total cholesterol, TG: Triglycerides; LDL-C: low density lipoprotein-cholesterol, HDL-C: high density lipoprotein-cholesterol; *n*: Sample size.

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
