# Peer review of "Hypolipidemic Effect of Arthrospira (Spirulina) maxima Supplementation and a Systematic Physical Exercise Program in Overweight and Obese Men: A Double-Blind, Randomized, and Crossover Controlled Trial"

_marinedrugs, 2019, doi:10.3390/md17050270_

Round 1
Reviewer 1 Report
Overall, the manuscript is very interesting, and is improving in quality during the review process.
In the figure legends, only statistical information are given for panel Fig 1E, Fig 2E and Fig 3A. The information of Dunn´s test should be indicated for the remaining panels.
If the Kruskal-Wallies test is not significant, the following posthoc test results can not be considered. For example, the test outcome for overweight participants for BMI is not significant for Kruskal-Wallies, but group specific differences are shown in the figure. This has to be corrected. Same logic applies for ANOVA and Tukey. For this variable, both tests were performed. Both not significant. Please correct.
Regarding the statistical tests done, the adjusted significance level has to be chosen to indicate the statistically significant difference between groups. At the moment, the authors have chosen the non-adjusted p-values, which is not correct. Multiple pairwise comparisons as in the chosen test, needs a Bonferroni-adjustment or the SPSS adjusted significance value. Please correct all panels in the figures and use the adjusted-significance level as indicators for statistical differences between treatments. For example, Fig 1A would be a/c/b/ab, Fig 1B would be a/b/b/b (as is), Fig 1C would be a/b/b/ab, and so on. Correct for all three figures dealing with differences of treatments. In the text, all related descriptions and conclusions should be adapted to those final statistically significant differences between treatment groups.
Further improvements can be done to the figures.
- use the same scale in the panels of each figure for better comparison.
- add "titles" above the panels of each figure for better perception (e.g. overweight, obese, dyslipidemic)
Edits to the text:
Abstract
- Change the second inserted phrase (in red) into: Comparing the final vs the initial values, BMI, total cholesterol, triglycerides and low-density lipoprotein cholesterol were decreased. High-density lipoprotein cholesterol increased in all treatment groups compared to the control.
2.2
- Change the inserted phrase (red) into: ... by Ex only in LDC-C considering all participants (overweight+obesity). Analysing dyslipidemic participants only, SE presented an improvement on all analysed parameter of blood lipids (p<0.05), Ex on ...
Author Response
Reviewer 1
We truly appreciate your contributions. The changes requested can be identified in the new version of our manuscript (marinedrugs-497435-R1) and the point-by-point response to your comments below:
· Overall, the manuscript is very interesting, and is improving in quality during the review process.
R= We appreciate your comment on this new version of the manuscript
· In the figure legends, only statistical information are given for panel Fig 1E, Fig 2E and Fig 3A. The information of Dunn´s test should be indicated for the remaining panels.
R= The change in the aforementioned sections has been made accordingly.
· If the Kruskal-Wallies test is not significant, the following posthoc test results cannot be considered. For example, the test outcome for overweight participants for BMI is not significant for Kruskal-Wallies, but group specific differences are shown in the figure. This has to be corrected. Same logic applies for ANOVA and Tukey. For this variable, both tests were performed. Both not significant. Please correct.
R= Thank you for showing us the error in figure 3. The change has been made according to your suggestion.
· Regarding the statistical tests done, the adjusted significance level has to be chosen to indicate the statistically significant difference between groups. At the moment, the authors have chosen the non-adjusted p-values, which is not correct. Multiple pairwise comparisons as in the chosen test, needs a Bonferroni-adjustment or the SPSS adjusted significance value. Please correct all panels in the figures and use the adjusted-significance level as indicators for statistical differences between treatments. For example, Fig 1A would be a/c/b/ab, Fig 1B would be a/b/b/b (as is), Fig 1C would be a/b/b/ab, and so on. Correct for all three figures dealing with differences of treatments. In the text, all related descriptions and conclusions should be adapted to those final statistically significant differences between treatment groups.
R= Thanks for this particular observation. We have rearranged all panels in all Figures as suggested, using the Bonferroni adjustment for multiple comparisons which is described in Section 4.9 as follows: “When any variable did not show a normal distribution (p<0.05) and when variances between groups were different, the Δ was analyzed by Kruskal-Wallis H and Dunn’s Post-hoc with Bonferroni adjustment for multiple comparisons”. Also, we have adapted the statistical differences in the text description and conclusions.
· Further improvements can be done to the figures. Use the same scale in the panels of each figure for better comparison. Add "titles" above the panels of each figure for better perception (e.g. overweight, obese, dyslipidemic)
R= All changes have been made accordingly.
· Edits to the text: (Abstract) - Change the second inserted phrase (in red) into: Comparing the final vs the initial values, BMI, total cholesterol, triglycerides and low-density lipoprotein cholesterol were decreased. High-density lipoprotein cholesterol increased in all treatment groups compared to the control.
R= The abstract has been reconstructed according to your suggestion
· Edits to the text: (2.2)- Change the inserted phrase (red) into: ... by Ex only in LDC-C considering all participants (overweight+obesity). Analysing dyslipidemic participants only, SE presented an improvement on all analysed parameter of blood lipids (p<0.05), Ex on ...
R= Section 2.2 has been improved due according to your suggestion and the new version of our manuscript has been reviewed and reedited by a Native English Speaker in order to improve its readability and quality. English editing changes (grammar and syntax) are highlighted in blue along the manuscript; we hope having answered what your requested.
Reviewer 2 Report
Revised manuscript is much improved. I appeciate your effort.
This study is consisted of 2 periods of 6 week-intervention and 2 weeks of wash out between. I think that all the data shown in Table 2 and Figures are combined average of the changes during 2 periods. 'Basal' in Table 2 shows the average of the blood test at day 0 (12 or 14 subjects)and day 56 (another 12 or 14 sujects). Also, 'Final' shows the average test data at day 42 and day 98? I consider that 'Changes' in Figures show the differences between day 42 and day0 and day 98 and day 56.
I think that the changes in blood lipids and body weight after the first and second intervention should be different, because the conditions of sujects at the starting point of day 0 and day 56 is not the same even though they've had washing out time. I am suggesting that you should show more detailed data at day 0, 42(first period), 56 and 98(second period) to confirm that doubt .
Author Response
We truly appreciate your contributions. The changes requested can be identified in the new version of our manuscript (marinedrugs-497435-R1) and the point-by-point response to your comments below:
· Revised manuscript is much improved. I appreciate your effort.
R= We appreciate your comment on this new version of the manuscript
· This study is consisted of 2 periods of 6 week-intervention and 2 weeks of wash out between. I think that all the data shown in Table 2 and Figures are combined average of the changes during 2 periods. 'Basal' in Table 2 shows the average of the blood test at day 0 (12 or 14 subjects) and day 56 (another 12 or 14 sujects). Also, 'Final' shows the average test data at day 42 and day 98? I consider that 'Changes' in Figures show the differences between day 42 and day 0 and day 98 and day 56. I think that the changes in blood lipids and body weight after the first and second intervention should be different, because the conditions of subjects at the starting point of day 0 and day 56 is not the same even though they've had washing out time. I am suggesting that you should show more detailed data at day 0, 42(first period), 56 and 98(second period) to confirm that doubt.
R= Effectively, Box-plots reported in Figures 1 to 3 show changes (Δ) in plasma lipid levels and BMI between day 42 vs. day 0, and day 98 vs. day 56, and we considered the possibility of differences of the conditions of the subjects between day 0 and day 56, but at do the statistical analysis, “basal” variables didn’t show differences, the analysis is shown below and was added to the Supplementary File 3, and to clarify the doubt, more information has been added to section 2.1 “Baseline characteristics of subjects at day 0 and 56 (wash-out period) didn’t show differences (p<0.05; see Supplementary File 3).

Dear authors,
Thank you for considering my comments on your manuscript.
I have read the manuscript in accordance with your responses to my comments. I believe you have responded well to some of them.
One of my concerns about your manuscript is that there are still many grammatical errors and, thus, a thorough proof-read of your manuscript is required.
Author Response
Reviewer 3
We truly appreciate your contributions. The changes requested can be identified in the new version of our manuscript (marinedrugs-497435-R1) and the point-by-point response to your comments below:
· Dear authors, thank you for considering my comments on your manuscript. I have read the manuscript in accordance with your responses to my comments. I believe you have responded well to some of them. One of my concerns about your manuscript is that there are still many grammatical errors and, thus, a thorough proof-read of your manuscript is required.
R= The new version of our manuscript has been reviewed and reedited by a Native English Speaker in order to improve its readability and quality. English editing changes (grammar and syntax) are highlighted in blue along the manuscript; we hope having answered what your requested.
Sincerely,
Dr. Arnulfo Ramos-Jiménez, Corresponding author
This manuscript is a resubmission of an earlier submission. The following is a list of the peer review reports and author responses from that submission.
Round 1
Reviewer 1 Report
Experimental design flow chart is not clearly understood and somewhat confusing.
Authors mentioned that participants decided if they belong or not to physical excercise intervention. If so, is it that each subject was allocated randomly only into two different groups with or without spirulina supplementation, not into 4 different groups ?
Have the subject wanted to be in exercise intervention gr initially exercised at first and second trials for 12 weeks in total?
In text, it is addressed that Table 2 shows the difference of blood lipid profile between day 0 and day 98. However, I know that each subject was allocated in one group at the first trial and then exchaged with the counter group after 2 week-wash out. Blood test was done at day 1, 42, 56 and 98. Would you explain in more detail how you use these blood data from the 4 time of tests to assess the difference in each group.
Do you think 2 weeks are enough in humans to wash out the biological effects of spirulina intake? You should suggest the supporting data or any evidences?
I expected that BMI will decrease in SE group and/or Ex group. But, it did not change in this study. Would you explain or discuss about these results?
Author Response
Response to Reviewer 1 Comments
Thank you for the time devoted to our article entitled: Hypolipidemic Effect of Arthrospira (Spirulina) maxima Supplementation and a Systematic Physical Exercise Program in Overweight and Obese Men: A Double-Blind, Randomized, and Crossover Controlled Trial, authored by Marco Antonio Hernández-Lepe, Abraham Wall-Medrano, José Alberto López-Díaz, Marco Antonio Juárez-Oropeza, Rosa Patricia Hernández-Torres, and Arnulfo Ramos-Jiménez (corresponding author). We have made all corrections in the new version of our manuscript (marinedrugs-456830-R1) according to your suggestions. Corrections and modifications were highlighted in red along the manuscript and a point-by-point response is described below.
· Experimental design flow chart is not clearly understood and somewhat confusing.
R= Experimental design flow chart (Figure 4) has been redesigned to show as clearly as possible the clinical trial characteristics.
· Authors mentioned that participants decided if they belong or not to physical excercise intervention. If so, is it that each subject was allocated randomly only into two different groups with or without spirulina supplementation, not into 4 different groups ?
R= Indeed, the randomization was only to divide the participants in the supplementation interventions (Spirulina maxima or placebo), we explained it in a clearer way in section 4.4 “To avoid study desertion, eligible participants (n= 52) decided if they belong or not to the systematic physical exercise program (where they stayed during the two treatments of the clinical trial), then they were randomly allocated to one of two possible supplementation interventions (S. maxima or placebo) divided in four treatments, including physical exercise program with S. maxima (SE) supplementation, physical exercise with placebo (Ex) supplementation, S. maxima supplementation without physical exercise program (Sm), or placebo supplementation without physical exercise program (C). The crossover was conducted only for the supplementation interventions”
· Have the subject wanted to be in exercise intervention gr initially exercised at first and second trials for 12 weeks in total?
R= All the subjects decided if they belonged to the group with systematic physical exercise program, where they stayed during the two treatments (12 weeks in total), the only intervention change that was made in the middle of the clinical trial was the placebo or Spirulina maxima supplementation, it has been clarified in the previous statement.
· In text, it is addressed that Table 2 shows the difference of blood lipid profile between day 0 and day 98. However, I know that each subject was allocated in one group at the first trial and then exchaged with the counter group after 2 week-wash out. Blood test was done at day 1, 42, 56 and 98. Would you explain in more detail how you use these blood data from the 4 time of tests to assess the difference in each group.
R= We appreciate your comment, because thanks to that we have noticed the error committed in such statement, so it has been edited in the text referring to that point being as follows “After the 42 days of all treatments, there existed statistical improve (p<0.05) of the blood lipid profile only by treatments with systematic exercise program in all the participants (overweight + obesity), but all intervention treatments presented a significant improve (p<0.05) of the blood lipid profile of dyslipidemic participants.”
· Do you think 2 weeks are enough in humans to wash out the biological effects of spirulina intake? You should suggest the supporting data or any evidences?
R= More information has been added about the wash-out period duration in section 4.4, including the following statement “Due to the lack of information in the literature, the durations for both, wash-out period (2 weeks) and treatment period (6 weeks) were considered long enough according to a systematic review of clinical trials that used Spirulina as treatment.”
· I expected that BMI will decrease in SE group and/or Ex group. But, it did not change in this study. Would you explain or discuss about these results?
R= Thanks for your comment, a paragraph of discussion focused on that point has been added “The greater synergistic effect of S. maxima and systematic physical exercise in all the variables studied observed in the obesity group compared with the overweight one, were not only due their higher baseline BMI and their blood lipids levels were mostly of dyslipidemia; the fact that the participants with obesity demonstrate significant reduction in BMI by all treatments and overweight participants did not show statistical difference in any treatment compared with control group is not surprising, given that in obesity there exists a lower activity of oxidative enzymes, resulting in an elevated intramuscular lipid content, mostly TG; for that reason, the capacity of obese subjects to use easily TG-derived fatty acids and circulating LDL-C as fuel during physical activity resulted in a higher lipid oxidation, which can be corroborated with the higher decrease of the blood lipid profile and the BMI in obesity”
Sincerely,
Dr. Arnulfo Ramos-Jiménez, Corresponding author
Reviewer 2 Report
The purpose of this study was to examine the effect of spirulina (S) ingestion and exercise (E) on lipid profiles of overweight and obese men. The results showed that a combination of SE or ingestion of S alone or exercise alone decreased total cholesterol and LDL and increased HDL.
Abstract:
1. If it is not restricted by the word limit (Journal style/format), the absolute values (pre and post) and probability levels (not just p<0.05) should be presented for each group.
2. There was inconsistency in terms of the frequency of exercise; were the exercise frequencies 2x/week or 5x week? In the methods section, exercise consisted of 3x/week of aerobic exercise and 2x/week of interval exercise.
Introduction:
1. The rational for the use of a combination of aerobic exercise and interval exercise should be included in the introduction
2. Has the spirulina been used in conjunction with aerobic exercise only or interval training only?
3. The rationale for using 6 weeks duration of exercise should be included in the introduction
Results:
1. The difference between the four groups should be presented; not just the change for each group.
2. Paired t-test can only be used to see the change for each group itself. However, the analysis would not show the difference of change scores between the four groups.
3. One-way ANOVA on change/delta scores should be used. Some of the change scores were similar, thus with one way ANOVA, results would be different. For example, just by examining the changes in HDL for overweight and obese participants (Fig 1). There were no differences in HDL change among SE, Ex, SM (for overweight) and Ex, Sm, C (for obese); the SE obese group HDL’s standard deviation was big.
4. The exact p values and the effect sizes should be presented.
Discussion:
1. The interaction between both aerobic exercise and interval exercise and spirulina should be discussed.
Methods:
1. It was not clear whether the cross-over was conducted within the exercise group only (exercise with or without spirulina/placebo) and the non-exercise group (spirulina /placebo) or between groups (e.g., the exercise group also acted as a non-exercise).
2. What types of exercise were involved in the muscular endurance exercise for 20-30 minutes?
3. What was the protocol for the interval exercise?
4. Did all participants comply with 5 days/week of exercise with no rest-day in between? Were all exercise sessions supervised?
5. Why was HR reserved used? What were exercise heart rates for both steady state exercise and interval exercise?
6. Why was maximal oxygen uptake not assessed?
7. Was 2 weeks of wash-out period following 6 weeks of each intervention sufficient?
8. Statistical analysis: paired t-test was not the appropriate analysis for 4 groups of participants.
Author Response
Thank you for the time devoted to our article entitled: Hypolipidemic Effect of Arthrospira (Spirulina) maxima Supplementation and a Systematic Physical Exercise Program in Overweight and Obese Men: A Double-Blind, Randomized, and Crossover Controlled Trial, authored by Marco Antonio Hernández-Lepe, Abraham Wall-Medrano, José Alberto López-Díaz, Marco Antonio Juárez-Oropeza, Rosa Patricia Hernández-Torres, and Arnulfo Ramos-Jiménez (corresponding author). We have made all corrections in the new version of our manuscript (marinedrugs-456830-R1) according to your suggestions. Corrections and modifications were highlighted in red along the manuscript and a point-by-point response is described below.
Abstract:
1. If it is not restricted by the word limit (Journal style/format), the absolute values (pre and post) and probability levels (not just p<0.05) should be presented for each group.
2. There was inconsistency in terms of the frequency of exercise; were the exercise frequencies 2x/week or 5x week? In the methods section, exercise consisted of 3x/week of aerobic exercise and 2x/week of interval exercise.
R= Several changes have been made in the abstract, but we could not add all your recommendations due we already have the limit of 200 words, thanks for your comments.
Introduction:
1. The rational for the use of a combination of aerobic exercise and interval exercise should be included in the introduction
R= More information has been added about the rationale for the use of aerobic exercise and interval exercise in the introduction, including the following statement: “many of the usual physical exercise programs prescribed involve only moderated aerobic exercise, although the scientific evidence of its effectivity as part of a weight loss or fat loss program alone is not clear, but at using high-intensity training as part of a systematic exercise program, (studies from only two weeks of 20 minutes sessions per week), numerous clinical benefits for both, healthy and CVD risk populations, like fat loss and blood lipids improvement have been proved.”
2. Has the spirulina been used in conjunction with aerobic exercise only or interval training only?
R= There is no information about clinical trials using spirulina together with aerobic exercise or high-intensity interval training, we have added the next statement in the text “There is no information in the literature about clinical trials using Spirulina together with aerobic exercise or high-intensity interval training, and only a few studies have focused on Spirulina supplementation effects against blood lipid profile of patients presenting different CVD risk factors”
3. The rationale for using 6 weeks duration of exercise should be included in the introduction
R= Thanks for your observation, and due we have to justify exercise and Spirulina treatment duration, more information has been added about the rationale for the use and duration of aerobic exercise and interval exercise in the introduction (Previously added) and about Spirulina in section 4.4 “Due to the lack of information in the literature, the durations for both, wash-out period (2 weeks) and treatment period (6 weeks) were considered long enough according to a systematic review of clinical trials that used Spirulina as treatment.”
Results:
1. The difference between the four groups should be presented; not just the change for each group.
R= We changed the meaning of the table so that the differences between groups could be observed, however as mentioned in the text, there were no differences between treatments, so we did not add the effect-sizes.
2. Paired t-test can only be used to see the change for each group itself. However, the analysis would not show the difference of change scores between the four groups.
R= Thanks for your comments, the description of the statistical analysis was corrected.
3. One-way ANOVA on change/delta scores should be used. Some of the change scores were similar, thus with one way ANOVA, results would be different. For example, just by examining the changes in HDL for overweight and obese participants (Fig 1). There were no differences in HDL change among SE, Ex, SM (for overweight) and Ex, Sm, C (for obese); the SE obese group HDL’s standard deviation was big.
R= The delta values did not show normality or homoscedasticity, so these analyzes were nonparametric; also, statistical analyzes were run again and some errors in the figures were corrected. For a better comprehension, brief information of the statistical analyzes made at the bottom of tables and figures is added.
4. The exact p values and the effect sizes should be presented.
R= Thanks for your comments, p values have been added to Table 2, but according to the data analyses, there did not exist significant effect size among treatment groups.
Discussion:
1. The interaction between both aerobic exercise and interval exercise and spirulina should be discussed.
R= Two new paragraphs have been added in the discussion section focused in your recommendation “Actual research focused in physical activity has found that the benefits of aerobic exercise are due to changes in the physical structure of blood carrier cholesterol proteins, resulting in an improvement of lipid levels, specifically, systematic exercise increases activity of hormone-sensitive lipase that hydrolyzes blood TG, then lipoprotein lipase acts on lipoprotein particles passing through the capillaries, releasing free fatty acids that may be taken up by skeletal muscle to regenerate damaged muscle fibers or esterified in phospholipids and intramuscular lipids or oxidized in the mitochondria; Nikolaidis et al. suggest that this degradation of endogenous lipids causes a shrinkage of lipoprotein particles, inducing the transference of the exceeding produced lipid layers to HDL-C.”
The greater synergistic effect of S. maxima and systematic physical exercise in all the variables studied observed in the obesity group compared with the overweight one, were not only due their higher baseline BMI and their blood lipids levels were mostly of dyslipidemia; the fact that the participants with obesity demonstrate significant reduction in BMI by all treatments and overweight participants did not show statistical difference in any treatment compared with control group is not surprising, given that in obesity there exists a lower activity of oxidative enzymes, resulting in an elevated intramuscular lipid content, mostly TG; for that reason, the capacity of obese subjects to use easily TG-derived fatty acids and circulating LDL-C as fuel during physical activity resulted in a higher lipid oxidation, which can be corroborated with the higher decrease of the blood lipid profile and the BMI in obesity”
Methods:
1. It was not clear whether the cross-over was conducted within the exercise group only (exercise with or without spirulina/placebo) and the non-exercise group (spirulina /placebo) or between groups (e.g., the exercise group also acted as a non-exercise).
R= We appreciate your comment, because thanks to that we have noticed the possible confusion, so it has been edited in the text referring to that point and Figure 2 has been redesigned.
2. What types of exercise were involved in the muscular endurance exercise for 20-30 minutes?
R= We already specified the types of exercise involved in the muscular endurance training, thanks for your observation “Muscular endurance exercise consisted in working all muscle groups (arms, legs, chest, back, and shoulder) one time a week, using a medium resistance-medium repetition protocol divided in four different exercises for each specific muscle group, doing three sets of 12-16 repetitions”
3. What was the protocol for the interval exercise?
R= According to your recommendation, we already explained the protocol for the interval exercise “The protocol for the high-intensity interval training consisted of 5 to 7 sets of 1 min of running at 80-90% heart rate reserve followed by 3 min of active resting at 50-70% heart rate reserve”
4. Did all participants comply with 5 days/week of exercise with no rest-day in between? Were all exercise sessions supervised?
R= We already add the observed points in section 4.2 “The elimination criteria were attendance by the subject of < 80% to the physical exercise sessions”, and section 4.7 “Subjects performed the physical exercise program in the UACJ gym, always under the technical supervision of a personal trainer.”
5. Why was HR reserved used? What were exercise heart rates for both steady state exercise and interval exercise?
R= Rationale to use HR reserve has been added to section 4.7 “Heart rate reserve was monitored due it is closely related to the percent of oxygen uptake reserve, making it easier to verify exercise intensity in the exercise prescription program”
6. Why was maximal oxygen uptake not assessed?
R= The effect of S. maxima and a systematic exercise program on the maximal oxygen uptake was assessed, those results have been previously published (Hernández-Lepe, M.; López-Díaz, J.; Juárez-Oropeza, M.; Hernández-Torres, R.P.; Wall-Medrano, A.;, Ramos-Jiménez, A. Effect of Arthrospira (Spirulina) maxima Supplementation and a Systematic Physical Exercise Program on the Body Composition and Cardiorespiratory Fitness of Overweight or Obese Subjects: A Double-Blind, Randomized, and Crossover Controlled Trial. Mar. Drugs. 2018, 16, 364. DOI: 10.3390/md16100364)
7. Was 2 weeks of wash-out period following 6 weeks of each intervention sufficient?
R= Information about the wash-out period and treatment period duration has been added in section 4.4, including the following statement “Due the lack of information in the literature, the durations for both, wash-out period (2 weeks) and treatment period (6 weeks) were considered long enough according a systematic review of clinical trials that used Spirulina as treatment”
8. Statistical analysis: paired t-test was not the appropriate analysis for 4 groups of participants.
R= The statistical analyses have been redesigned and are explained in section 4.9, including the following statement “In order to analyze statistical differences among treatments and time, univariate repeated measures ANOVA designs with two within-subjects (initial and final values), and four inter-subjects (treatments) factors were used”
Sincerely,
Dr. Arnulfo Ramos-Jiménez, Corresponding author
Reviewer 3 Report
The authors conducted a randomised, double-blind, crossover, controlled trial to investigate whether Arthrospira (Spirulina) maxima (S. maxima) supplementation with and without a systematic physical exercise program affected blood lipid levels in 52 overweight and obese men. The authors hypothesised that S. maxima intake with or without a systematic exercise program would lower levels of total cholesterol (TC), triglycerides (TG), and low-density lipoproteins-cholesterol (LDL-C) but increase high-density lipoproteins-cholesterol (HDL-C) levels in both overweight and obese men. Men were, first, divided into two groups; 1) systematic physical exercise group and 2) non-exercise group, then, randomised using a 1:1 allocation to receive either the 4.5g daily supplement or placebo for 6 weeks. After a two-week washout period, each group switched treatments (i.e. supplementation to placebo and vice versa). Outcomes reported were daily energy intake, blood lipid profile by treatment group and by cases with dyslipidemia, and body mass index. The results showed that supplementation with exercise did reduce TC and LDL-C, and an increase in HDL-C for the supplementation with exercise group for both overweight and obese men at follow-up compared to baseline. Similar results were also found in cases with dyslipidemia, although a significant decrease in LDL-C levels was also found. BMI significantly reduced in the SE and Sm groups compared to controls for obese men.
Overall, I thought this paper was well organised, interesting, and suitable for publication in Marine Drugs. The article describes a well-executed trial and appropriate for the respective conditions (overweight/obesity). No participants dropped out during the trial, which is one of its strengths. The authors have published the trial protocol in BMJ Open, which I referred to whilst writing my review and I did not spot any major deviations. Nevertheless, I have a few comments that would help strengthen the quality of this article. I have divided my comments into major and minor comments below.
Major comments:
Results:
Baseline characteristics showed that there were significantly lower TC and LDL-C levels in overweight men compared to obese men. This is somewhat expected, but not really discussed anywhere in the article. The change in TC and LDL-C levels were larger in obese men and this change may have simply been due to these men having much higher levels at baseline. It would be helpful for this to be discussed a little in the article.
Discussion section:
I think it would be helpful for the authors to state clearly in the first paragraph whether their hypothesis was fully, partially, and not supported by the results of the trial. I would say that the hypothesis is partially supported by the results as not all predicted outcomes were observed.
This section could elaborate more on the implications of the findings. For example, in what ways could the results of this trial inform other similar trials? What would be the clinical implications of the trial?
Methods:
The article states that men “decided if they belong or not to [a] physical exercise intervention.” This implies that there is allocation bias as men are choosing to receive either the physical exercise program or not. This might be a misunderstanding from the way this sentence is written. If so, it would be beneficial for the authors to clarify this process by rephrasing this sentence.
In section 4.6, it states that the dietary records were inspected for missing data. However, the statistical analysis section does not mention how missing data was handled. It is possible that no missing data was identified. If this is the case, a line somewhere in the methods or results section is needed to clarify this.
Minor comments:
The article could do with further proof-reading. There were many grammatical errors throughout the article. For example, in the first paragraph there is the following sentence “…being obesity and low physical activity two of the most important factors” would be rewritten as ‘…being obese and less physically active are two of the most important factors.’
Table 1 – I think presenting the means and SDs would suffice. I do not think the 95% CIs are necessary and would make the table clearer.
Table 2 – I think using asterisk (*) instead of bold numbers would be better for clarity. I would also include Ns for dyslipidemic men in the table as well. Again, I would remove 95% CIs and just report means and SDs.
When reporting results for Figures 1-3, I would report the means and CIs instead of SDs for consistency between the text and figures.
In section 2.4, the first sentence states that all treatments had a statistically significant effect on BMI for overweight men. However, I believe this refers to changes between baseline to follow-up within subjects. If so, this sentence would need to be rewritten to clarify this because Figure 3 shows there are no significant effects on BMI between treatments.
In section 2.4, it states that BMI were “statistically related” to decreases in blood lipid profiles. I would rephrase with ‘statistically correlated’ to avoid the reader assuming causality.
I hope the authors find my comments helpful and I wish them good luck with this article.
Author Response
Response to Reviewer 3 Comments
Thank you for the time devoted to our article entitled: Hypolipidemic Effect of Arthrospira (Spirulina) maxima Supplementation and a Systematic Physical Exercise Program in Overweight and Obese Men: A Double-Blind, Randomized, and Crossover Controlled Trial, authored by Marco Antonio Hernández-Lepe, Abraham Wall-Medrano, José Alberto López-Díaz, Marco Antonio Juárez-Oropeza, Rosa Patricia Hernández-Torres, and Arnulfo Ramos-Jiménez (corresponding author). We have made all corrections in the new version of our manuscript (marinedrugs-456830-R1) according to your suggestions. Corrections and modifications were highlighted in red along the manuscript and a point-by-point response is described below.
· Major comments:
· Results:
· Baseline characteristics showed that there were significantly lower TC and LDL-C levels in overweight men compared to obese men. This is somewhat expected, but not really discussed anywhere in the article. The change in TC and LDL-C levels were larger in obese men and this change may have simply been due to these men having much higher levels at baseline. It would be helpful for this to be discussed a little in the article.
R= We appreciate your comment, it has been considered at adding the following paragraph in the discussion section “The greater synergistic effect of S. maxima and systematic physical exercise in all the variables studied observed in the obesity group compared with the overweight one, were not only due their higher baseline BMI and their blood lipids levels were mostly of dyslipidemia; the fact that the participants with obesity demonstrate significant reduction in BMI by all treatments and overweight participants did not show statistical difference in any treatment compared with control group is not surprising, given that in obesity there exists a lower activity of oxidative enzymes, resulting in an elevated intramuscular lipid content, mostly TG; for that reason, the capacity of obese subjects to use easily TG-derived fatty acids and circulating LDL-C as fuel during physical activity resulted in a higher lipid oxidation, which can be corroborated with the higher decrease of the blood lipid profile and the BMI in obesity”
· Discussion section:
· I think it would be helpful for the authors to state clearly in the first paragraph whether their hypothesis was fully, partially, and not supported by the results of the trial. I would say that the hypothesis is partially supported by the results as not all predicted outcomes were observed.
R= We already clarify how our hypothesis was only partially supported, thanks for your comment “Our hypothesis is partially supported by the results of the clinical trial due to the fact that all the predicted outcomes were not observed since there was a significant improvement in all the dependent variables of all the participants, but not in distinct groups of overweight and obesity.”
This section could elaborate more on the implications of the findings. For example, in what ways could the results of this trial inform other similar trials? What would be the clinical implications of the trial?
R= We have added some paragraphs in the Discussion section focused in clarify the clinical implications of our study, including the following statements “Many of the beneficial effects of Spirulina are attributed to its nutritional content, but its action mechanisms are poorly understood; our findings corroborate previous investigations in animal models; some authors suggest that a possible component responsible of the S. maxima hypolipidemic effect is a protein named C-phycocyanin, which reduce the blood lipid profile; Iwata et al. studied the effect of a diet containing Spirulina compared to a high fructose diet in rats, they reported a statistical decrease in TG concentration after the intervention, and attribute the possible action mechanism to the metabolism of lipoproteins, due that rats fed with the diet with Spirulina showed a significant increase in the activity of lipoprotein lipase; other authors suggest that C-phycocyanin increases endogenous enzymes activity, scavenging free radicals, and downregulates cofactors in fat metabolism like the reduced form of nicotinamide adenine dinucleotide phosphate;. Nagaoka et al. attribute Spirulina hypolipidemic effects to the fact that dietary supplementation with the cyanobacteria seems to decrease the intestinal assimilation of cholesterol, probably because Spirulina compounds bind to bile acids in the jejunum, affecting the micellar solubility of cholesterol and then suppresses cholesterol absorption. These biological processes might explain the underlying mechanisms of S. maxima involved in the significant improvement on TC, TG, and lipoprotein-associated cholesterol found in this research, but appropriate clinical trials are needed to elucidate this.”
· Methods:
· The article states that men “decided if they belong or not to [a] physical exercise intervention.” This implies that there is allocation bias as men are choosing to receive either the physical exercise program or not. This might be a misunderstanding from the way this sentence is written. If so, it would be beneficial for the authors to clarify this process by rephrasing this sentence.
R= We have redesigned Figure 4 and rewrite the mentioned sentence “To avoid study desertion, eligible participants (n= 52) decided if they belong or not to the systematic physical exercise program (where they stayed during the two treatments of the clinical trial), then they were randomly allocated to one of two possible supplementation interventions (S. maxima or placebo).”
· In section 4.6, it states that the dietary records were inspected for missing data. However, the statistical analysis section does not mention how missing data was handled. It is possible that no missing data was identified. If this is the case, a line somewhere in the methods or results section is needed to clarify this.
R= In the last paragraph of the discussion we have added the strength points of the study, including your excellent observation, thank you very much “The strong points of the present study were that no missing data were identified, no participants dropped out during the trial”
· Minor comments:
· The article could do with further proof-reading. There were many grammatical errors throughout the article. For example, in the first paragraph there is the following sentence “…being obesity and low physical activity two of the most important factors” would be rewritten as ‘…being obese and less physically active are two of the most important factors.’
R= The new version of our manuscript has been reviewed carefully to correct grammatical errors in order to improve its readability and quality, and we have considered your specific recommendation.
· Table 1 – I think presenting the means and SDs would suffice. I do not think the 95% CIs are necessary and would make the table clearer.
R= We have followed your recommendation, and effectively, Table 1 is clearer without the 95% CIs.
· Table 2 – I think using asterisk (*) instead of bold numbers would be better for clarity. I would also include Ns for dyslipidemic men in the table as well. Again, I would remove 95% CIs and just report means and SDs.
R= We have redesigned Table 2 according your recommendations, we really appreciate all your comments to improve our manuscript quality.
· When reporting results for Figures 1-3, I would report the means and CIs instead of SDs for consistency between the text and figures.
R= According to your recommendation, we already report 95% CI in text and Figures, thank you.
· In section 2.4, the first sentence states that all treatments had a statistically significant effect on BMI for overweight men. However, I believe this refers to changes between baseline to follow-up within subjects. If so, this sentence would need to be rewritten to clarify this because Figure 3 shows there are no significant effects on BMI between treatments.
R= Thanks for your observation, we already corrected the sentence as follows “According to Figure 3, no treatment showed a significant effect (p<0.05) against BMI (kg·m−2) in the overweight group”
· In section 2.4, it states that BMI were “statistically related” to decreases in blood lipid profiles. I would rephrase with ‘statistically correlated’ to avoid the reader assuming causality.
R= We appreciate your recommendation, which has been added to section 2.4 “Decreases in BMI were statistically correlated to decreases in blood lipid profile, to TC with r= 0.49 (p<0.01), to TG with r= 0.22 (p<0.05), and to LDL-C with r= 0.42 (p<0.01).”< span="">
Sincerely,
Dr. Arnulfo Ramos-Jiménez, Corresponding author